# Optical properties and composition of viscous organic particles found in the Southern Great Plains

Matthew Fraund[1], Daniel J. Bonanno[1], Swarup China[2], Don Q. Pham[1], Daniel Veghte[3], Johannes Weis[4,5,a], Gourihar Kulkarni[2], Ken Teske[6], Mary K. Gilles[4], Alexander Laskin[2,b], Ryan C. Moffet [1,c]

1. Department of Chemistry, University of the Pacific, Stockton, California, 95204, USA
2. Environmental and Molecular Sciences Laboratory, Pacific Northwest National Laboratory (PNNL), Richland, Washington 99352, USA
3. Center for Electron Microscopy and Analysis, The Ohio State University, Columbus, Ohio, 43212, USA
4. Chemical Sciences Division, Lawrence Berkeley National Laboratory, Berkeley, California, 94720, USA
5. Department of Chemistry, University of California, Berkeley, California, 94720, USA
6. Atmospheric Radiation Monitoring (Southern Great Plains Climate Research Facility), Billings, Oklahoma, 74630 USA
a. Current address: Physikalisches Institut, Universität Würzburg, Am Hubland, 97074 Würzburg, Germany
b. Current address: Chemistry Department, Purdue University, West Lafayette, Indiana, 47907, USA
c. Current address: Sonoma Technology, Petaluma, California, 94954, USA

*Correspondence to:* Matthew Fraund (m_fraund@u.pacific.edu)

**Abstract.** Atmospheric high viscosity organic particles (HVOP) were observed in samples of ambient aerosols collected on April and May 2016 in the Southern Great Plains of the United States. These particles were apportioned as either airborne soil organic particles (ASOP) or tar balls (TB) from biomass burning based on spetro-microscopic imaging and assessment of meteorological records of smoke and precipitation data. Regardless of their apportionment, the number fractions of HVOP were positively correlated ($R^2 = 0.85$) with increased values of Ångström absorption exponent (AAE) measured *in-situ* for ambient aerosol at the site. Extending this correlation to 100% HVOP yields an AAE of 2.6, similar to previous literature reports of the class of light absorbing organic particles known as brown carbon (BrC). One out of the three samples investigated had a significant number of ASOP while the other two samples contained TB. Although there are chemical similarities between ASOP and TB, they can be distinguished based on composition inferred from near edge absorption X-ray fine structure (NEXAFS) spectroscopy. ASOP were distinguished from TB based on their average –COOH/C=C and –COOH/COH peak ratios, with ASOP having lower ratios. NEXAFS spectra of filtered soil organic brine particles nebulized from field samples of standing water deposited after rain were consistent with ASOP when laboratory particles were generated by bubble bursting at the air-organic brine interface. However, particles generated by nebulizing the bulk volume of soil organic brine had particle composition different from ASOP. These observations are consistent with the raindrop generation mechanism responsible for ASOP emissions in the area of study. In contrast, nebulized samples carry with them higher fractions soil inorganics dissolved in the bulk volume of soil brine that are not aerosolized by the raindrop mechanism. Our results support the bubble bursting mechanism of particle generation during rainfall resulting in the ejection of soil organics into the atmosphere. In addition, our results show that ASOP may only be atmospherically relevant during times when suitable emission conditions are met.

## 1 Introduction:

Regional and global atmospheric transport models are commonly used to predict the impact of aerosols on radiative forcing (Feng et al., 2013). The efficacy of these models relies on estimates of the types, number concentrations, spatial distribution, and emission sources of aerosols. One challenge of continued scientific discussion is how (and to what extent) industry and other anthropogenic activities contribute to climate forcing. Emissions of soot, one of the most well-studied anthropogenic aerosols emitted by fossil fuel combustion, have been shown to have a strong climate warming factor comparable to carbon dioxide (Bond et al., 2013). However, soot is not the only light absorbing carbon containing aerosol of concern. Less absorbing, but often more abundant light absorbing organic carbon aerosol, known as brown carbon (BrC), has been the subject of increased investigation, with studies showing that BrC may account for a substantial fraction of the total aerosol radiative forcing (Feng et al., 2013;Zhang et al., 2017). These particles exhibit a wavelength dependent light absorption, absorbing stronger at shorter wavelengths, giving them a brown appearance. Ongoing research on the role of BrC in global radiative forcing suggests it is insufficiently represented in models (Feng et al., 2013;Laskin et al., 2015;Bond and Bergstrom, 2006).

BrC particles are defined in very general terms of their chemical composition and optical properties (Bond et al., 2013;Laskin et al., 2015). BrC refers to a broad category of organic particles comprised of many different chromophores originating from a variety of sources (Andreae and Gelencsér, 2006). Sources of BrC include, but are not limited to: biomass burning (Rizzo et al., 2013;Laskin et al., 2015), biogenic fungi, humic like substances (Andreae and Gelencsér, 2006;Wang et al., 2016), and secondary organics (Laskin et al., 2015). One example of BrC particles are tar balls (TB) which are most commonly found in biomass burning emissions downwind of smoldering fires (Chakrabarty et al., 2010). TB of 50-300 nm in size have a characteristic spherical morphology observed in many studies, indicative of their highly viscous or "glassy" phase state. Recently, larger spherical organic particles (~500 nm) of a different BrC type (Veghte et al., 2017) have been also observed in rural Oklahoma, with their composition consistent with soil organic matter (SOM). These newly discovered BrC particles have been termed airborne soil organic particles (ASOP) (Wang et al., 2016). The study reported formation of highly viscous submicron particles following the impaction of rain droplets on the soil surfaces which eject ASOP into the ambient air (Wang et al., 2016). Mechanism of plausible ASOP emissions was further corroborated by laboratory experiments with water droplets falling onto surfaces of soil proxies. Briefly, as falling water droplets make contact with the porous soil surface, air from within the soil is trapped beneath the resulting water layer. Bubbles form as the droplet sinks and mixes with the soil. These bubbles then rise to the air-water interface bringing dissolved soil organics with them where a cavity forms and then ruptures producing aerosols containing compounds entrained from the soil (Joung and Buie, 2015).

A defining characteristic of BrC particles is the spectral dependence of light absorption in the visible wavelengths. One way of quantifying this dependence is to measure the absorption coefficient ($\sigma_{ap}$) as a function of wavelength, to calculate the absorption Ångström exponent (AAE) (Backman et al., 2014). The AAE value is essentially the slope of a log-log plot of absorption coefficient and wavelength, with higher values corresponding to enhanced absorption in shorter wavelengths.

Because soot absorbs nearly equally across the entire visible spectrum, it has an AAE value of approximately 1. In contrast, reported AAE values for BrC are substantially higher, typically in a range of 2.5-6 (Rizzo et al., 2013;Lawless et al.,
2004;Hoffer et al., 2006) but could be as high as 9-11 (Andreae and Gelencsér, 2006;Lin et al., 2017). An AAE value of 2.5 has been used as a lower limit to attribute absorption, at least in part, to BrC (Lack and Langridge, 2013). In addition to the AAE, absorption due to BrC has also been investigated through calculation of the complex refractive index (Veghte et al., 2017).

To better characterize highly viscous organic particles (HVOP) appearing as solid spheres, samples of ambient
particles were collected at the atmospheric radiation measurement (ARM) facility in Lamont, Oklahoma located in the southern great plains (SGP) as part of the Holistic Interactions of Shallow Clouds, Aerosols, and Land-Ecosystems (HI-SCALE) field campaign (Fast et al., 2019). To compare the spectroscopic signatures of atmospheric particles with SOM typical for the area of study, aquatic samples of the surface layer of muddy puddles were also collected around the sampling site. These aquatic samples were then filtered, nebulized, and the resulting particles were impacted onto microscopy substrates. The purpose of
collecting these samples was to use them to compare particle morphology and composition when AAE was high, as indicated by online measurements performed at the site (Springston et al., 2016). The particle samples were analyzed with both scanning electron microscopy (SEM) and scanning transmission X-ray microscopy coupled with near-edge X-ray absorption fine structure (STXM-NEXAFS) spectroscopy. SEM images were taken at a 75° tilted angle with respect to the surface normal to identify HVOP suspected to be either ASOP or TB. STXM was then used to obtain chemical images of particles at the C K-
edge spectral range to distinguish ASOP and TB based on their carbon NEXAFS spectra (Wang et al., 2016).

The present work evaluates the appearance of ASOP following rain events and determines their particle-specific spectroscopic characteristics that would enable distinguishing them from biomass burning TB, SOA, or other anthropogenic sources (Parworth et al., 2015;Sheridan et al., 2001). Characterizing the properties and emission sources of unaccounted BrC particles like ASOP is vital to ensure the proper representation of BrC in local and regional climate models. Multiple
experimental records (both real-time and spectro-microscopic) were used here to assess the underreported ASOP. Using these techniques in tandem with longstanding atmospheric measurements will also help in advancing our knowledge of how ASOPs fit into the broader class of BrC and how ASOP may affect radiative forcing by aerosols.

## 2 Experimental:

### 2.1 Sample Collection

Samples of atmospheric particles were collected at the ARM SGP field site located in north central Oklahoma [36° 36' 18.0" N, 97° 29' 6.0" W], at an altitude of 320 meters (Ferrare et al., 2006). Sampling was performed before forecasted and after observed rain events during the rainy season from April 26th through May 17th, during day and night periods separately. Within this timeframe, samples were collected on 17 days with each sampling period beginning between 8:00 to 21:00 local time. The total sampling time was approximately eight hours (unless interrupted by a major rain event) with

samplers operated using a 30/30 min on/ off duty cycle. Similarly, 4 nighttime samples were collected between the hours of 18:00 and 6:00 local time with a duty cycle of 60/30 min on/off. Rain events interrupted two sampling days (May 8th and May 16th) in which no sampling took place. The primary objective was to observe particle types during sunny days following major rain events, where a threshold precipitation rate of 10 mm/hr was used to define a rain event, when enhanced ASOP concentration would be likely. Note that ASOP may be present during some periods without a recent local rain event due to

transport from elsewhere.

Particles were collected by impaction using a Micro-Orifice Uniform Deposit Impactor (MOUDI, MSP 100) attached to a rotating motor that rotates the eight impaction surfaces (known as stages) to facilitate uniform particle deposition. The impactor was connected to a ¾ max horsepower (0.56 kW) vacuum pump (General Electric Motors & Industrial Systems, 10 PSI rating). With a 30 L/min sampling flow rate, the micro-orifice nozzles reduce jet velocity, pressure drop, particle bounce

and re-entrainment. This MOUDI was connected to a mesh covered sampling inlet which was tilted downward, avoiding unwanted collection of descending debris, insects, and other sources of contamination, and was positioned ~6 meters above the ground.

Samples for analysis were selected from two stages with the following particle size cutoff ranges: stage 7 (0.32 to 0.56 μm) and stage 8 (0.18 to 0.32 μm). These stages were chosen to cover part of the size ranges for TB (50 to 300 nm) (Tóth

et al., 2014) and ASOP (300 to 800 nm) (Wang et al., 2016) while also constraining the number of samples to analyze. Substrates of $Si_3N_4$ film supported by a silicon wafer (0.5 X 0.5 mm$^2$ $Si_3N_4$ window, 100 nm membrane thickness, 5 X 5 mm$^2$ Si frame; Silson, Inc.) and filmed TEM copper grids (Carbon type B film, Copper 400 mesh grids; Ted Pella, Inc.) were used as impaction substrates.

In addition to impaction samples of ambient particles, aquatic samples containing SOM brine were collected in 50-

200 mL aliquots via syringes from mud puddles surrounding the SGP field site. This was performed to evaluate a connection between organics from terrestrial aquatic samples, the hypothesized source of ASOP, and ambient ASOP. Four SOM aquatic samples were collected on May 17th 2016, prior to offline analysis of microscopy samples. The obtained samples were then nebulized in laboratory experiments using a Collison nebulizer (3 jet MRE, CH Technologies USA) and collected on stage 8 of a 10-stage impactor (110-R, MSP, Inc.) to produce 300-500 nm diameter particles, where subsequent tilted SEM imaging

revealed that ASOP comprised up to 80% of the particles by number. In addition, an aliquot of 30-40 mL of one SOM sample was used to generate particles by bubbling $N_2$ gas (at 8 L/min) through the liquid using a fritted glass bubbler. The $N_2$ gas, now carrying particles generated from the bursting of bubbles at the air-water interface, was directed into a cascade impactor (Sioutas Personal Cascade Impactor #225-370, SKC) where particles were collected on a pre-loaded microscopy grid to mimic the hypothesized mechanism of ASOP formation. Stage D of the Sioutas impactor was used ($D_{50}$ 0.25 μm) to best correspond

with the size ranges of MOUDI samples taken

**2.2 STXM Measurements and Image Processing.**

The STXM instruments (beamline 5.3.2.2 ALS, Berkeley, CA, USA, and the SM1 beamline CLS, Saskatoon, SK, Canada) used in this work are located in the Advanced Light Source (ALS) at the Lawrence Berkley National Laboratory and at the Canadian Light Source (CLS) at the university of Saskatchewan (Kilcoyne et al., 2003;Regier et al., 2007). Breifly, monochromatic soft X-rays are focused down to a spot size ranging from 20 to 40 nm in diameter. The sample is raster scanned after a region of sufficient particle concentration is found and individual images are captured at selected photon energis. Maps were collected in addition to spectra, which are images consisting of 8 energies around the elemental absorption k-edges, four energies corresponding to C absorption at 278.0 eV, 285.4 eV, 288.6 eV, and 320.0 eV, two for N absorption at 398.0 eV and 430.0 eV, and 2 for O absorption at 525.0 eV and 550.0 eV. These energies are used to identify and characterize the basic chemical composition maps of particles on a pixel-by-pixel basis (Fraund et al., 2017). The two C energies at 285.4 and 288.6 eV are used for the identification of soot or elemental carbon (EC). The absorption peak at 285.4 eV occurs due to the excitation of the C 1s $\rightarrow \pi^*_{C=C}$ transition (* indicating the excited state) which is indicative of $sp^2$ hybridized carbon-carbon bonds (C=C). This excitation of $sp^2$ hybridized carbon is prominent for soot (Bond and Bergstrom, 2006). To identify soot, the intensity of the $sp^2$ peak (relative to the pre-edge at 278 eV after normalization to 320 eV) must be equal to or exceed 35% of that of highly ordered pyrolytic graphite (Moffet et al., 2010). Absorption at 288.6 eV is representative of the C 1s $\rightarrow \pi^*_{R(C^*=O)OH}$ transition characteristic of carboxylic acid groups (COOH) that are very common in atmospheric organic carbon. These photon energies aid in differentiating individual particles based on the molecular speciation of carbon, making this method convenient for analysis of the field collected particles (Moffet et al., 2010). Spatial displacement between images within a stack does occur and, whenever needed, it has been corrected for by utilizing the image registration algorithm developed by Guizar-Sicarios' (Fraund et al., 2017;Guizar-Sicairos et al., 2008).

**2.3 SEM Measurements**

The SEM analysis of particle samples was performed at the Environmental Molecular Sciences Laboratory (EMSL) located at Pacific Northwest National Laboratory (PNNL). Particles were imaged using a computer-controlled scanning electron microscope (FEI, Quanta 3D FEG, Hillsboro, AL, USA). SEM images were initially taken orthogonally to the substrate until a particle laden region on the substrate was identified. The substrate mount was then tilted by 75° in order to identify spherical particles. Based on the tilted images, particles which had an aspect ratio (height divided by width) greater than 0.8 were identified as spherical HVOP. The SEM images were categorized based on aspect ratio to determine the fraction of HVOP (high aspect ratio) compared to flat, non-spherical particles (low aspect ratio) for ensembles of approximately 400 total particles. This counting was performed manually for both ambient and lab-generated samples.

## 2.4 PSAP Measurements.

The Particle Soot Absorption Photometer (3-λ PSAP) instrument measures light transmission through aerosol filter samples at three wavelengths: red (660 nm), green (522 nm) and blue (470 nm) (Springston, 2018). PSAP measurement taken at the SGP ARM facility is a standard data product (ARM, 2019). Equation (1) shows how absorption coefficients $\sigma_{ap}$ are calculated from raw PSAP data using spot size ($A$), sample volume ($V$), and average filter transmittances for incident ($I_O$) and transmitted ($e$) light through particle laden filter.

$$\sigma_{ap} = \frac{A}{V}\ln\left[\frac{I_o}{I}\right] \tag{1}$$

Implementing corrections specified in Bond *et al.* (1999), as shown in Eq. (2), minimizes noise present in the PSAP data, which is a result of inherent unit-to-unit variability in field instrumentation. Additionally, these corrections mitigate systematic error from filter loading. The absorption reported by the PSAP instrument ($\sigma_{PSAP}$) includes an inherent calibration for a given measurement period which monitors the change in transmission by using the previous sample as a blank.

$$\sigma_{PSAP} = \frac{\sigma_{ap}}{2(0.5398\tau + 0.355)} \tag{2}$$

The raw absorption coefficient of a sample ($\sigma_{ap}$) at a given time is normalized by filter transmission ($\tau$), which is reset after the installation of a new filter ($\tau = 1$ for an unloaded filter) (Springston, 2018). The corrected absorption coefficients were used to calculate the *AAE* values. Backman *et al.* (2014) introduced the following equation relating Ångström Exponent (*AE*) to the wavelength ($\lambda$) dependent extinction (Backman et al., 2014):

$$\ln\left[\sigma_E(\lambda)\right] = -AE\ln[\lambda] + C \tag{3}$$

here, $\sigma_E$ refers to the extinction coefficient at wavelength $\lambda$. While the *AE* represents the spectral dependence of combined scattering and absorption, the *AAE* is specific to absorption and is obtained by substituting $\sigma_{ap}$ for $\sigma_E$ (Backman et al., 2014). By taking Eq. (3) at a given wavelength and subtracting Eq. (1) at another wavelength, the constant C can be removed, resulting in a more practical equation:

$$AAE = \frac{-\ln\left[\frac{\sigma_{ap}(\lambda_1)}{\sigma_{ap}(\lambda_2)}\right]}{\ln\left[\frac{\lambda_1}{\lambda_2}\right]} \tag{4}$$

Using Eq. (4), PSAP data was used to calculate the *AAE* of ambient aerosols. Appearance of BrC was then inferred from PSAP time records based on elevated values of *AAE* >1.7 (Kirchstetter et al., 2004).

# 3 Results:

## 3.1 Optical Properties of High Viscosity Organic Particles

Spherical HVOP have similar spectral characteristics to BrC regardless of their origin. Both TB and ASOP show characteristic increased absorption for shorter wavelengths of visible light compared to longer wavelengths (Alexander et al., 2008;Veghte et al., 2017). During periods when they are prevalent, overall aerosol optical properties should start to resemble those of BrC. To investigate the presence of BrC with this wavelength dependence, PSAP data was utilized to determine time records of AAE as shown in Fig. 1 alongside the corrected absorption coefficients. Time records of CO mixing ratios and particle number concentrations are also shown in Fig. 1 to provide further information on the composition of air mass. Figure 1 shows how both red and blue absorption coefficients and AAE change over the course of this study. Rain events were defined as periods of time where rainfall exceeded 10 mm/hr. If two rain events were observed less than 30 minutes apart, they were considered one event. The vertical gray bars show sampling periods when aerosol with BrC properties was detected and which particle samples were imaged with STXM and SEM.

Both the absorption coefficient and AAE time series data were collected with minute time resolution; the data has been averaged over 30-minute time windows to emphasize longer term data trends instead of short-term fluctuations. The AAE sometimes shows an increase after rain events, though it is not consistent; because aerosol production from rain is more complex than rainfall amounts (depending on droplet size and impact velocity as well as soil characteristics) it is difficult to see a direct correlation between rain and AAE or particle concentration. While ASOP emissions are expected during rain events, precipitation scavenging is also occurring (Joung and Buie, 2015). The net effect of these two competing processes likely depends on many environmental conditions and is not yet clear. During a rain event, ASOP will contribute to an increased AAE but because AAE is a bulk optical property the presence of other absorbing aerosols like black carbon or the washout of mineral dust (large particles with a high AAE) can dampen the effect that rain events have on AAE (Bergstrom et al., 2007).

From this time record, a few samples stand out: the night of April 28th, May 5th, and May 14th as these samples had elevated AAE >1.7. On May 5th and April 28th measured AAE values were above 2, warranting analysis of particle samples (Lack and Langridge, 2013). While the May 14th sample doesn't show particularly high AAE values, compared to the entire time series, it was collected after a heavy rain storm passed through the area. Also, the particle concentration immediately following the rain event on May 14th shows a significant level of precipitation scavenging, reducing the number of background particles present during sampling, concentrating any ASOP produced (Hegg et al., 2011). Of note, the lower the $\sigma_{ap}$ drops, due to rain or otherwise, the more pronounced the effect PSAP noise has on the calculated AAE. In addition to having high AAE values, both the April 28th and May 5th periods showed elevated particle and CO concentrations indicating plausible anthropogenic activity or biomass burning plumes. Figure S1 shows the time series of trace gasses (CO, $O_3$ and $SO_2$) and particle concentration which can be used to identify anthropogenically influenced air plumes. In addition to elevated CO levels, the April 28th and May 5th samples show elevated $SO_2$ concentrations while some of the highest $O_3$ concentrations seen over

the sampling periods were observed during May 5th. In contrast, the May 14th period exhibits low or fairly average concentrations of these trace gasses.

## 3.2 Chemical Imaging of High Viscosity Organic Particles

Bulk optical properties, like an elevated AAE, may suggest the presence of spherical HVOP, therefore these measurements were used to select samples for detailed chemical imaging of particles. First, tilted SEM images were taken and HVOP fractions were observed in individual samples, ranging from 5% to near 70% by number. Figure 2 shows representative microscopy and spectro-microscopy images for three days where HVOP fractions were high. The top row shows the tilted SEM images used to identify HVOP. Magenta arrows point to a few identified HVOP to highlight how much they stick out above the substrate compared to the others. Of note, the SEM images also show presence of what looks like fractal soot particles in the April 28th and May 5th samples.

Second, the same samples were later imaged by the STXM spectro-microscopy. The middle and bottom row of images shown in Fig. 2 are STXM chemical speciation maps and total carbon absorbance (TCA) maps, respectively. Following the procedure described in Moffet (2010) for chemical speciation maps, each pixel is assigned as either inorganic dominant, organic dominant, or as a region with high C=C bonding (Moffet et al., 2010). The TCA maps indicate the thickness of each of the particles (as calculated using previously published thickness equations (Fraund et al., 2019;O'Brien et al., 2015)) normalized by the individual area equivalent diameter. Values close to 0 represent flat particles, while values closer to 1 represent taller, nearly spherical particles. As was suggested in the SEM images, soot is present in both the April 28th and May 5th samples, while the May 14th sample has only organic and inorganic particles. In the April 28th and May 5th samples, circular soot-containing particles are associated with the highest TCA. In contrast, the May 14th sample contained high TCA spherical particles comprised of organic dominant material only.

Sample collection information from the 7 samples where the fractions of HVOP were calculated is presented in Table 1 below. The highest HVOP fraction was observed with the samples taken on May 5th. The prevalence of these particles can be seen in Fig. 2 in the top row. Elevated fractions of HVOP were also found for the April 28th night sample taken at 18:30 and for the May 14th sample. While the April 28th and May 5th sample showed elevated HVOP fractions for both stage 7 and 8 samples, the May 14th sample is unique in that a higher HVOP fraction was only found for the smaller stage. In addition, the April 28th and the May 5th sample both have elevated particle concentrations and CO mixing ratios, suggesting more polluted conditions possibly due to anthropogenic activity or biomass burning events. In contrast, the time period corresponding to May 14th sample shows the lowest particle concentration and CO mixing ratio. Also of note is that the April 28th and May 5th samples were both taken long after the last rain, with May 5th being taken multiple days afterwards whereas the May 14th sample was taken 10 hours after the last rain.

Optical properties of individual HVOP from this same data set have been investigated in our previous work (Veghte et al., 2017). There, the complex refractive index from 200 to 1200 nm was calculated for HVOP found in the April 28th sample using electron energy loss spectroscopy (EELS). The imaginary part (k) of the refractive index is related to light

absorption and can be related to the absorption coefficient using $\sigma_{ap} = 4\pi k/\lambda$ (Jennings et al., 1979). Absorption coefficients over the 200 – 1200 nm wavelength range were calculated from a published imaginary refractive index plot (Veghte et al., 2017). From this plot published in Veghte (2103), the AAE was calculated using $\sigma_{ap}$ values for 660 and 470 nm and a resulted in a value of 1.41. This is close to the value calculated in the present work (1.42, Table 1) for the April 28th sample (a difference of only 0.01) showing that the two methods agree, at least for one sample.

**Table 1. Ambient sampling information. Stage 8 values in parentheses where available.**

| Start Date (CDT)[1] | MOUDI Stage | Duration (hr) | HVOP (%) | AAE (Red/Blue)[2] | Hours Since Last Rain[3] | Particle Conc. (cm$^{-3}$)[4] | CO (ppb)[5] | # Particles Imaged SEM | # Particles Imaged STXM | HVOP Source |
|---|---|---|---|---|---|---|---|---|---|---|
| 26 Apr 14:00 | 7 | 1.5 | <5 | 0.808 | >72 | 2100 | 150 | 450 | 0 | Low HVOP |
| 28 Apr 9:45 | 7 | 5 | 3 | 1.20 | 30 | 4100 | 140 | 200 | 0 | Low HVOP |
| 28 Apr 18:30 | 7 (8) | 10 | 23 (25) | 1.42 | 39 | 10300 | 160 | 900 | 28 | Biomass Burning, TB |
| 1 May 11:30 | 7 (8) | 4 | 12 (15) | 1.23 | 45 | 1000 | 130 | 500 | 0 | Low HVOP |
| 2 May 20:00 | 7 | 10 | 13 | 1.15 | 80 | 1900 | 130 | 900 | 185 | Low HVOP |
| 5 May 8:00 | 7 (8) | 13 | 70 (60) | 2.01 | 140 | 6000 | 170 | 1000 | 32 (214) | Biomass Burning, TB |
| 14 May 11:00 | 7 (8) | 5 | 10 (35) | 1.29 | 10 | 650 | 120 | 300 | 50 (86) | Biogenic, ASOP |

[1]Central Daylight Time (UTC -5) [2]Particle Soot Absorption Photometer (PSAP) [3]Video Disdrometer [4]Condensation Particle Counter (CPC) [5]ARM/Aerosol Observatory System (AOS)

Even though elevated HVOP numbers were identified on a number of days, Fig. 1 shows that there is no clear relationship between the particle concentration and AAE. One reason for this is the presence of other absorbing or non-absorbing aerosols which will increase the measured particle concentration while affecting the AAE differently than HVOP are expected to. To address this, a correlation plot was made (using values found in Table 1) between the AAE values and the HVOP fractions and a strong correlation was found ($R^2 = 0.85$) as seen in Fig. S2. Extrapolating this linear correlation to 100% HVOP yields an AAE value of 2.6, consistent with previously reported AAE values for BrC (Lack and Langridge, 2013). This correlation also uses only aerosols impacted onto MOUDI stages 7 and 8, whose combined aerodynamic size range (0.18 – 0.56 µm) covers much of the ambient aerosol surface area distribution (Seinfeld and Pandis, 2006). Because optical properties like AAE will be most sensitive to changes in this size range, we exclude particles which are counted in the overall particle concentration but contribute less to bulk optical properties. This not only suggests that the HVOP found here are BrC, but that they warrant consideration by models due to their measureable effect on bulk aerosol-radiation interactions as they can occupy a significant fractions of the aerosol fine mode.

The appearance of viscous HVOP at the SGP site has been reported previously (Wang et al., 2016). There, they showed that the viscous HVOP at SGP had an elevated total carbon absorption (TCA, defined by the pre-edge OD subtracted from the post-edge OD) compared to other carbonaceous particles of similar sizes. High TCA values indicate particles that were not deformed upon impaction suggesting a high viscosity. Figure 3 shows TCA values of individual particles plotted against circular equivalent diameter (CED) for four samples reported in this work. The April 28th, May 5th, and May 14th samples all have elevated TCA whereas the May 2nd sample shows lower carbon absorption, in line with the TCA values

characteristic of lab-generated SOA particles (Wang et al., 2016). Note that the May 5[th] sample, where the highest HVOP fraction was identified, has the smallest particles with TCA values above the ambient organic particle regions. Contrast this with the May 2[nd] sample, which shows very few particles with high TCA values and a correspondingly low HVOP fraction.

So far, the above analysis applies to a general class of HVOP, which can include both ASOP and TB particles. Additional considerations are necessary, however, before any conclusions are tied to ASOP exclusively. The two samples,
with the highest HVOP fractions and AAE values, were those collected more than 39 hours after a rain event. Because emission of ASOP is associated with bursting of bubbles at flooded soil surfaces after rain, the HVOP found in the April 28[th] and May 5[th] samples are likely not locally emitted ASOP, while ASOP might be present in the sample of May 14[th].

To investigate the nature and source of HVOP and determine which can be confidently classified as ASOP, smoke and fire from biomass burning sources (NOAA:OSPO, 2019) and precipitation data (NWS, 2019), were used along with the
285 calculations of air backward trajectories using a hybrid single particle Lagrangian integrated trajectory model (HYSPLIT) (Stein et al., 2015;Rolph et al., 2017). This data for the events when the three samples had elevated HVOP fractions are shown in Fig. 4 below: Apr 28[th], May 5[th], and May 14[th]. Additional information about the HYSPLIT trajectory conditions, as well as trajectories calculated from multiple starting altitudes, is available in Fig. S3.

April 28[th] had a moderate fraction of HVOP (25 ± 0.6%) along with the second highest AAE over the sampling
periods studied here. This sample was also taken about 39 hours after the last rain, which makes ASOP less likely to be found. Figure 4 shows that the corresponding air mass trajectories passed over a few burning fires and while smoke was present in some of the fires surrounding the sampling site, the air mass did not pass through these regions. Precipitation data shows rainfall in some of the surrounding states but none close to the sampling site. The storm surrounding Oklahoma may have produced some ASOP which could have been transported to the sampling site, although elevated particle concentration and
slightly enhanced CO, $SO_4$, and $N_2O$ concentration at the time of sampling suggested biomass burning as more prominent emission source (Koppmann et al., 2005). Although it is possible ASOP may make up a fraction of the HVOP seen during the April 28[th] sample, immediately east of the sampling site are a coal-fired power plant, an oil refinery, and natural gas power plants which could possibly contribute to the HVOP seen on this day. Furthermore, as seen in Fig. 4 (as well as in more detail in Fig. S3), the April 28[th] sample has an initial eastward direction.

May 5[th] has the highest HVOP fraction and the highest average AAE but it also had been days (140 hours) since the last rain event, making ASOP unlikely. There were many fires surrounding the sampling site compared to the other sampling periods and the back trajectories show air masses passing directly over some of these fires suggesting the presence of associated smoke emissions. Precipitation data shows that no rainfall was observed anywhere near the sampling site. Figure 1 also shows that this sampling date coincided with a slight particle concentration enhancement and the highest CO mixing ratios observed
over this period of the field campaign. Figure S1 also shows enhancement in $O_3$ and $SO_2$ concentrations. Because TB are found within smoke plumes, the high HVOP fraction observed in the April 28[th] and May 5[th] samples may be primarily (or exclusively) due to TB (Pósfai et al., 2004). The presence of combustion byproducts like TB and soot is supported by the STXM carbon speciation maps shown in Fig. 2, where particles with elevated C=C bonding constituents are often attributed to soot.

The last sample date shown (May 14[th]) has regions of smoke away from the sampling site with backward trajectories heading from just outside the smoke filled region. However, because a rain event was recorded 10 hours prior to sampling where significant precipitation scavenging was observed (see the particle concentration decrease in Fig. 1), no influence from biomass burning was observed in this sample. The precipitation map shows that precipitation was observed over the sampling site as well as many of the surrounding areas (Radke et al., 1980). Because the microscopy samples were taken shortly after it had rained, the $35 \pm 1.3\%$ fraction of HVOP observed at that time may likely be related to ASOP, contributing to the spike in AAE seen during this sampling period.

With the HVOP observed in the microscopy images being around 0.6 μm in diameter, it is expected that they would be aloft and present for the sampling period for at least 10 hours after the rain event (Williams et al., 2002). Any particles of this size travelling from further away, such as ASOP produced elsewhere, through some of the surrounding storm are less likely to be scavenged due to the lowered precipitation scavenging efficiency (Greenfield scavenging gap) at approximately 0.1 μm (Radke et al., 1980). This could have effectively concentrated ASOP particles as the air mass travelled through the surrounding storm. It is also possible that emission conditions for ASOP (soil and rainfall characteristics) were ideal somewhere along the airmass trajectory. This would have the effect of bringing in a high concentration of ASOP after the aerosol plume traveled. In addition, production of viscous SOA particles was likely taking place at the same time (Virtanen et al., 2010). The peak in AAE seen during the May 14[th] sampling period occurs as the sun is rising and ozone concentration rapidly increases (this also coincides with an increase in particle concentration). Although viscous SOA particles would contribute to the AAE, they are formed <100 nm in size, limiting their effect on optical properties. The peak in AAE here, and in the other cases, drops down quickly due to the particle-laden airmass moving away from the sampling site.

The influence of smoke shown in Fig. 4 may account for the enhancement of HVOP fractions without rainfall in the April 28[th] and May 5[th] samples, likely due to TB. The carbon STXM/NEXAFS spectra of TB have been recorded previously and its characteristic features are shown in Fig. 5 (Tivanski et al., 2007). The same figure compares the STXM/NEXAFS spectra for both ambient particles collected during this study and lab generated ASOP proxies. The right panel of Fig. 5 also includes three characteristic spectra of HVOP particles from the April 28[th], May 5[th] and May 14[th] samples. Even though May 5[th] and April 28[th] had the highest AAE values and the highest HVOP fractions, many hours since the last rain along with the presence of smoke suggest they might be TB, consistent with their NEXAFS spectral features. Three apparent peaks are common for these spectra: the C=C peak at 285.3 eV, the COH peak at 286.7 eV, and the R–(C=O)OH peak at 288.6 eV, all of which are present in the previously reported TB spectra (Tivanski et al., 2007). May 5[th] spectra also show a small feature around 289.5 eV which is present (and more prominent) in the TB spectra, a peak which is associated with alkyl carbon bonded to oxygen, often alcohols. This similarity reinforces the assumption that May 5[th] contains a large amount (70%) of HVOP attributable to TB.

Upon comparison with the April 28[th] and May 5[th] samples, NEXAFS spectra of HVOP particles from the May 14[th] sample (taken 10 hours after raining) show a slightly enhanced C=C peak and an almost absent COH peak. These same features can be seen in the previously reported NEXAFS spectra of ASOP (Wang et al., 2016;Veghte et al., 2017) and "free light" SOM

isolated in that work for comparison. One reason for the difference in COH peak intensity may be due to the presence of levoglucosan or other plant derived products such as polysaccharides, tannins, or lignin fragments (Marín-Spiotta et al., 2008)

(which contains multiple –OH groups) in the samples affected by smoke plumes, a common product of biomass burning from the pyrolysis of carbohydrates (Lakshmanan and Hoelscher, 1970). Another differentiating factor is the ratio of intensities between the –COOH peak at 288.6 eV and the C=C peak at 285.3 eV. In the TB spectrum, the –COOH peak is much higher than the C=C peak compared to the ASOP spectrum (Wang et al., 2016) and this difference is borne out in the spectra collected for the current study as well.

The chemical composition of TB has been reported in literature, with fresh TB being comprised various biomass tar products with a substantial degree of aromaticity. The nonpolar products were the most strongly associated with the wavelength dependence in absorption seen in TB and were found in greater number in fresh TB (Li et al., 2019). Photochemical oxidation in the presence of $O_3$ or OH radicals was shown to bleach this wavelength dependence after about 3.5 days (Sumlin et al., 2017). Photooxidation was suggested to break network of conjugated double bonds in the TB chromophores, resulting in more

oxygenated (carbonyl substituted) products which concentrated on the surface of the TB (Hand et al., 2005;Tóth et al., 2014). ASOP are comprised, in part, of high molecular weight humic-like substances. These compounds contain multiple conjugated ring systems which likely serve as chromophores in a similar way to the tar materials in TB (Kumada, 1955). The extensive conjugated systems may be driving the enhanced C=C peak in Fig. 5 for ASOP associated spectra. Also, because humic-like substances in ASOP are substituted by many different functional groups, rather than the nonpolar components of TB, more

reactive sites may be available for oxidation. This may lead to faster atmospheric processing and bleaching of ASOP wavelength dependent absorption. This aging could also serve to increase the viscosity of ASOP in the same way that it does to TB (Adachi et al., 2019). Another factor differentiating their aging processes could be their hygroscopicity. Reports have shown TB change very little in morphology even when cycled from 0 to 100% RH, having a growth factor of ~1.09 (Adachi and Buseck, 2011;Semeniuk et al., 2007). However, because ASOP are formed via dissolved SOM that is ejected during

precipitation, they are expected to be more hygroscopic. Indeed, Wang (2016) showed the results of RH cycling in supplemental figures and found a growth factor of ~1.15 at 85% RH along with droplet activation at 98% RH.

Also shown is a spectrum of organic particles not associated with HVOP. This spectrum is characterized by small C=C and COH absorptions with an intense –COOH peak. The large –COOH peak seen in non-HVOP organics is indicative of much higher contributions of carboxylic functional groups which define the solubility of the organic particles. This increased

solubility would lower the viscosity of the non-viscous organic particles due to their substantial water content and thus these particles would be deformed and flatten upon impaction. Viscous SOA have been observed previously, under both laboratory and ambient conditions (Virtanen et al., 2010). The HVOP discussed in the current work differ from these viscous SOA by virtue of their mode of formation with both TB and ASOP being comprised of larger organic compounds. Furthermore, the particles themselves are larger. Viscous SOA are formed and observed as much smaller particles (<100 nm) then TB and

ASOP (300 – 700 nm).

The left-panel of Fig. 5 shows STXM/NEXAFS spectra of ASOP proxies generated from the SOM brine. The top four spectra from the puddle water samples all show a fairly strong carbonate signal at around 290.1 eV along with two broad potassium peaks ($L_2$ and $L_3$) at about 298 eV. Also, the corresponding inset carbon speciation map shows a large, inorganic dominant region (teal blue). The presence of large inorganic regions is not consistent with the mainly organic particles seen previously. However, because these samples were nebulized from bulk solutions to begin with, water soluble carbonates from the soil must have been present upon nebulization. To better model the bubble bursting mechanism of generating ASOP, dry $N_2$ gas was bubbled through puddle water samples and particles resulted from the bubble bursting at the air-solution interface were collected. In these experiments, generated particles showed almost a complete reduction of the carbonate peak and a small reduction in the potassium peaks, plus the carbon speciation map (Fig. 5 inset) showed an entirely organic dominant particle. Comparison between the spectra of ASOP proxies generated in the 'bubbling experiment' with the ambient spectra of HVOP from May 14th shown in the right panel indicates substantial similarities between their spectral features. This includes the diminished –COOH/C=C peak ratio and the relative absence of a COH peak.

For more quantitative comparison two sets of peak ratios were calculated. The first between the –COOH and C=C absorptions and another between the –COOH and COH peaks. The peak ratios are plotted in Fig. 6 below for all of the ambient samples, the proxy particles from the "bubbling" experiment, and the two literature spectra. As noted above, the TB spectra have higher –COOH/C=C and –COOH/COH peak ratios. The separation between these two ratios is also the largest for the TB spectrum. While the April 28th and May 5th peak ratios are not quite as high, they all bear a strong resemblance to the TB spectrum. Perhaps alone the similarity would not be enough to define these ambient samples as TB; however, coupled with the other data presented here the peak ratios support the HVOP seen in the May 5th sample as being TB. The April 28th peak ratios are further separated than the literature ASOP peak ratios are, but closer together than the TB literature peak ratios. With the possibility of ASOP travelling from surrounding states (with active rainfall) as well as the proximity of smoke plumes it is likely that the HVOP observed on April 28th are comprised of both TB and ASOP. The May 14th sample peak ratios however, are much different and are more comparable to the ASOP peak ratios. The –COOH/C=C ratio is much lower in both cases and the –COOH/COH ratio is of the same value. From the lab generated aerosols, the bubbling sample peak ratios are also similar to the ASOP peak ratios, suggesting that bubbling reproduces the mechanism of ASOP generation. Another noteworthy observation is the difference between the April 28th or May 5th peak ratios and the May 14th peak ratios. Although each of these samples showed the presence of HVOP and each sample's AAE suggested that these HVOP were BrC, there is a stark contrast between the smoke influenced samples (Apr 28th and May 5th) and the rainfall influence samples (May 14th), suggesting different sources of HVOP in these two cases.

## 4 Conclusion

BrC particles like TB and ASOP and their place in the global aerosol budget are yet insufficiently understood. Here, it was shown that presence of HVOP is correlated with the BrC properties of overall aerosols, indicated by the elevated values of AAE. On multiple days HVOP were observed to comprise a significant fraction of the fine mode aerosols.

Tilted SEM was used to identify HVOP fractions in a number of samples taken during this study, and the fractions of HVOP present for each sample were determined based on the aspect ratios of individual particles. The HVOP fractions showed a strong correlation with the average AAE over the sampling periods, with an $R^2$ of 0.85. When extrapolated to 100% fraction of HVOP an AAE of 2.6 was calculated, consistent with literature reported values of BrC (Lack and Langridge, 2013). These observed AAE values suggest that BrC relevant particles can be identified by methods of chemical imaging based on three dimensional morphology coupled with chemical composition. These samples were further classified into samples with TB and samples with ASOP based on their NEXAFS characteristics and by comparing smoke and precipitation data during corresponding collection periods.

Chemical imaging showing the differences between ASOP and TB laden samples was performed using STXM/NEXAFS spectromicroscopy. Samples unaffected by recent rain, collected while smoke plumes were present showed a higher –COOH/C=C peak ratio and an elevated –COH peak intensity. The elevated –COH peak is likely due to the presence of sugars such as levoglucosan or other less oxidized molecules. The sample from May 14th was collected 10 hours after a rain event and had less influence from smoke plumes. This sample showed a much more subdued –COH peak and a smaller –COOH/C=C peak ratio. Comparing the ambient spectra collected here with previously collected spectra supported the presence of TB in the smoke-affected samples and the presence of ASOP in the sample taken after a rain event. Peak ratios between –COOH and C=C and between –COOH and COH were calculated to serve as a quantitative metric that can be potentially used to differentiate between TB and ASOP, and, probably more generally, between the smoke-affected samples and the samples with particles induced by rainfall.

HVOP are a subclass of BrC particles which can define the nature of aerosol-radiation interactions during time periods when they are prevalent. Differentiating between types of HVOP like TB and ASOP has proven to be a challenging task that relies on subtle differences in chemical composition and atmospheric conditions at the time of sampling. Of the three sampling days focused on in this study, only one indicated an appreciable number of ASOP present. Because the conditions necessary for ASOP emission depend both on soil properties and precipitation characteristics, the dominant source of HVOP will often be TB due to the frequency of biomass burning instances along with the large number of particles they emit. ASOP are likely to contribute to aerosol properties, optical and otherwise, only during short time periods where the emission conditions are met. Further questions still exist about ASOP specifically. How do soil characteristics affect the composition of ASOP? How are ASOP transformed as they travel through the atmosphere? What are the emissions factors of ASOP? Answering questions like these may improve the quality of models in regions where large areas with open soils such as agricultural fields and grasslands are exposed to intensive rains, especially during rainy seasons when ASOP might be prevalent.

*Data availability*. The data set used here is available for download as a .zip file at https://doi.org/10.17605/OSF.IO/G8FPW
(Fraund, 2020).

*Code availability*. MatLab code used for the current work is available as a Supplement.

*Author contributions*. Conceptualization, M.F., D.J.B., S.C., D.V., M.K.G., A.L., and R.C.M.; Software, M.F., D.J.B, and
R.C.M.; Formal Analysis, M.F. and D.J.B.; Investigation, M.F., D.J.B., S.C., D.Q.P., D.V., J.W., G.K., and R.C.M.; Resources,
K.T., M.K.G., A.L., and R.C.M.; Data Curation, M.F., D.J.B., S.C., D.V., and R.C.M.; Writing – Original Draft, M.F. and
D.J.B.; Writing – Review & Editing, all authors; Visualization, M.F., D.J.B., and S.C.; Supervision, M.K.G., A.L., and R.C.M.
; Project Administration, M.K.G., A.L., R.C.M.; Funding Acquisition, M.K.G., A.L., and R.C.M.

*Competing interests*. The authors declare that they have no conflict of interest

*Acknowledgements*. We acknowledge support from the US Department of Energy's Atmospheric System Research program,
Office of Biological and Environmental Research (OBER), award DE-SC0018948. PSAP records obtained from the
Atmospheric Radiation Measurement (ARM) User Facility of OBER was used in this study. Chemical imaging of particles
was performed at The Advanced Light Source at Lawrence Berkeley National Laboratory. The beamline staff of 11.0.2 and
5.3.2 helped make the current work possible, specifically: David Shapiro, David Kilcoyne, Matthew Markus, and Hendrik
Ohldag. A portion of the research was performed using EMSL (grid.436923.9), a DOE Office of Science User Facility
sponsored by the Office of Biological and Environmental Research. Part of the research described in this paper was performed
at the Canadian Light Source, a national research facility of the University of Saskatchewan, which is supported by the Canada
Foundation for Innovation (CFI), the Natural Sciences and Engineering Research Council (NSERC), the National Research
Council (NRC), the Canadian Institutes of Health Research (CIHR), the Government of Saskatchewan, and the University of
Saskatchewan. The CLS work was done at the SM beamline 10ID-1 with help of its staff: Jian Wang, Yingshen Lu, and Jan
Geilhufe. The NOAA HYSPLIT transport and dispersion model (http://www.ready.noaa.gov) was used to calculate backward
trajectories for this publication. We would acknowledge Shih-Ming Huang from Sonoma Technology Inc. for suggesting the
use of the archived precipitation data from the National Weather Service's (NWS) Advanced Hydrologic Prediction Service
reported here for the selected periods of our study.

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

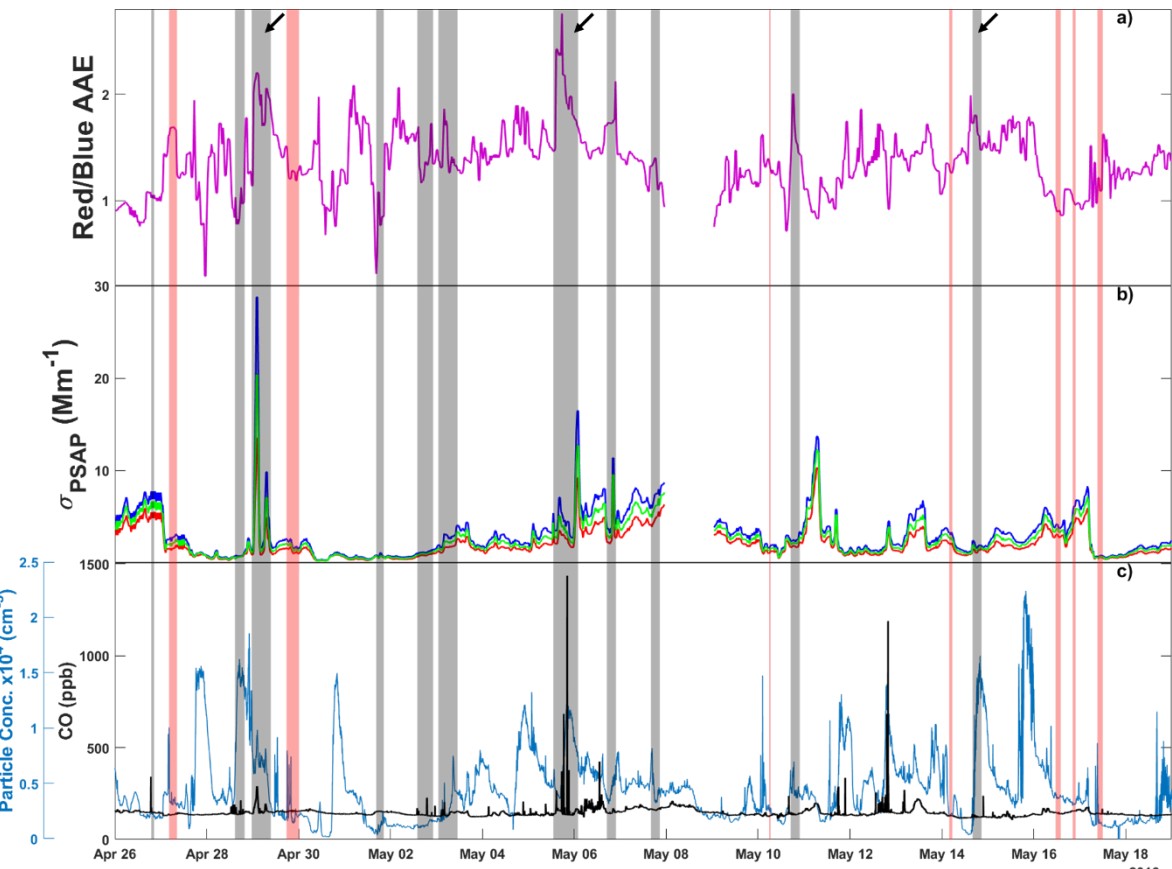

**Figure 1: Time series for a) absorption Ångström exponent (AAE) calculated from red and blue absorption coefficients, b) absorption coefficients measured at red (660 nm), green (522 nm), and blue (470 nm) wavelengths, and c) CO mixing ratio and ambient particle concentration. Grey vertical bars indicate sampling periods where microscopy samples were collected. Red vertical bars represent rain events. Arrows indicate the three sampling periods investigated in the current work where BrC particles were expected. PSAP data for May 8th is not available due to instrument error.**


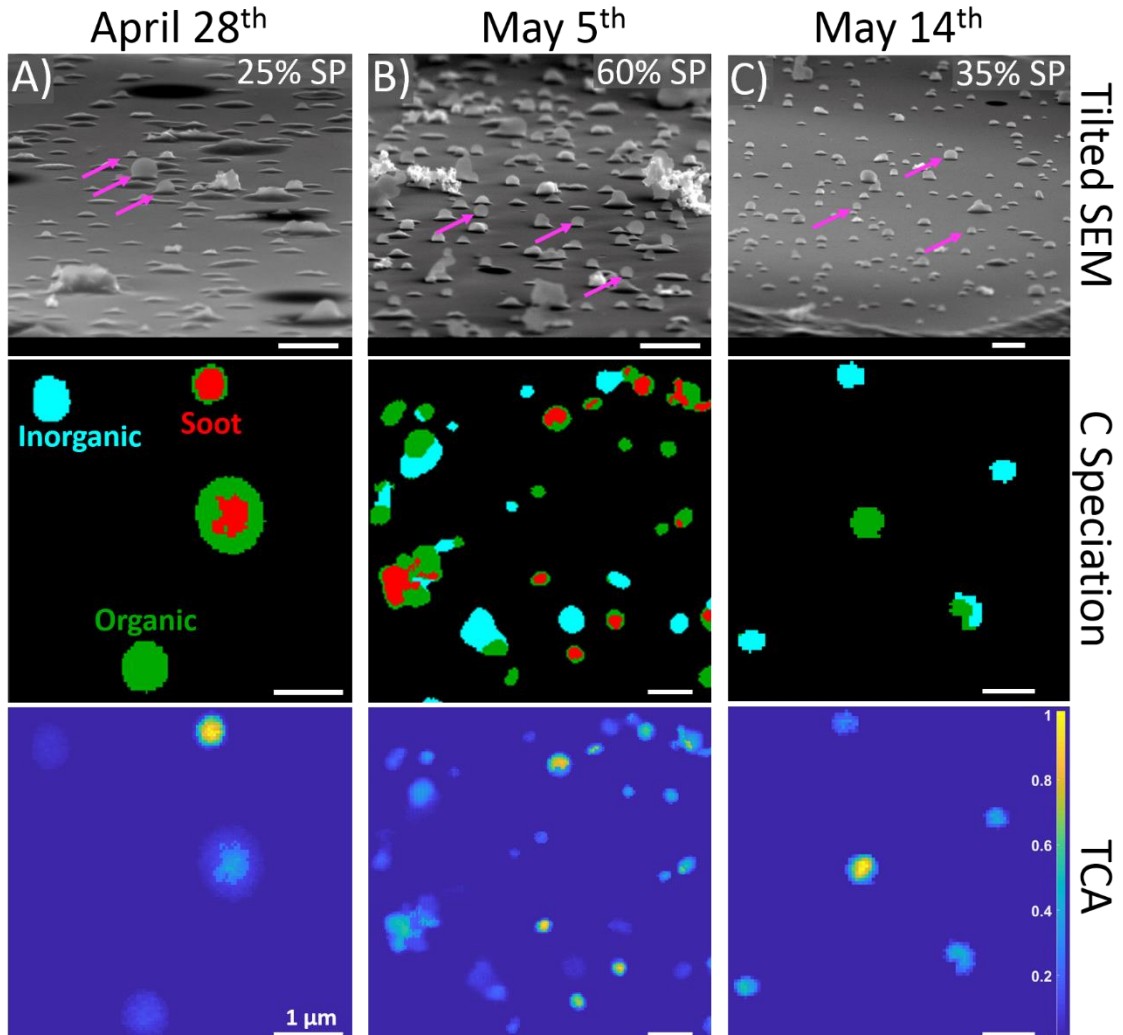

**Figure 2: (Top row)** Tilted (75°) SEM images show differences in HVOP fractions between three samples. Each of these samples were collected on a MOUDI stage 8, which selected for particles in the 150-360 nm size range. Magenta arrows point to characteristic HVOP. **(Middle row)** Carbon speciation maps with red representing regions with enhanced C=C bonding, green representing organics, and teal representing inorganics. **(Bottom row)** Total carbon absorption (TCA) images calculated from dividing thickness by the area equivalent diameter of each particle. The carbon speciation and TCA maps show the same fields of view. All scale bars are 1 μm in length

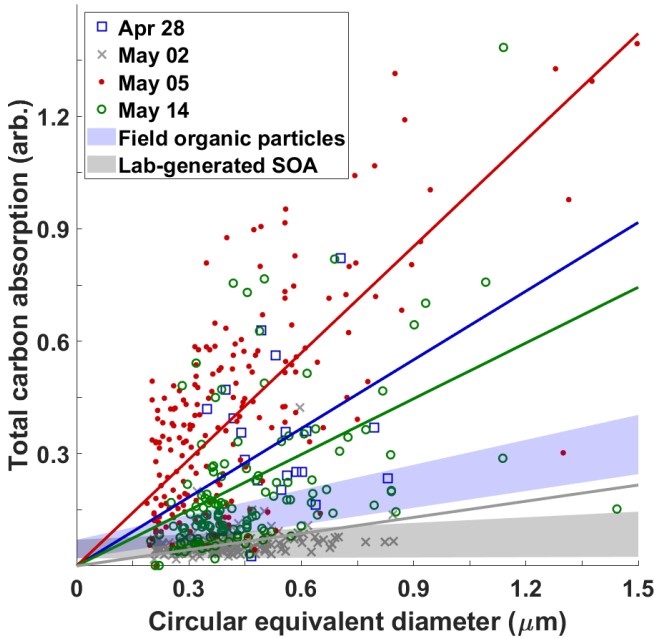

Figure 3: Correlation between total carbon absorption (TCA) and size measured by circular equivalent diameter for four sampling dates with colored best-fit lines (anchored at 0) for each sample. Blue and gray shaded regions show regions characteristic of ambient organic particles and lab-generated secondary organic aerosols reported in previous study (Wang et al., 2016). The sharp cutoff at about 0.2 µm is due to the selected detection limit of small particles (5 or fewer pixels in diameter) from raw data to avoid falsely identifying noise spikes as particles.

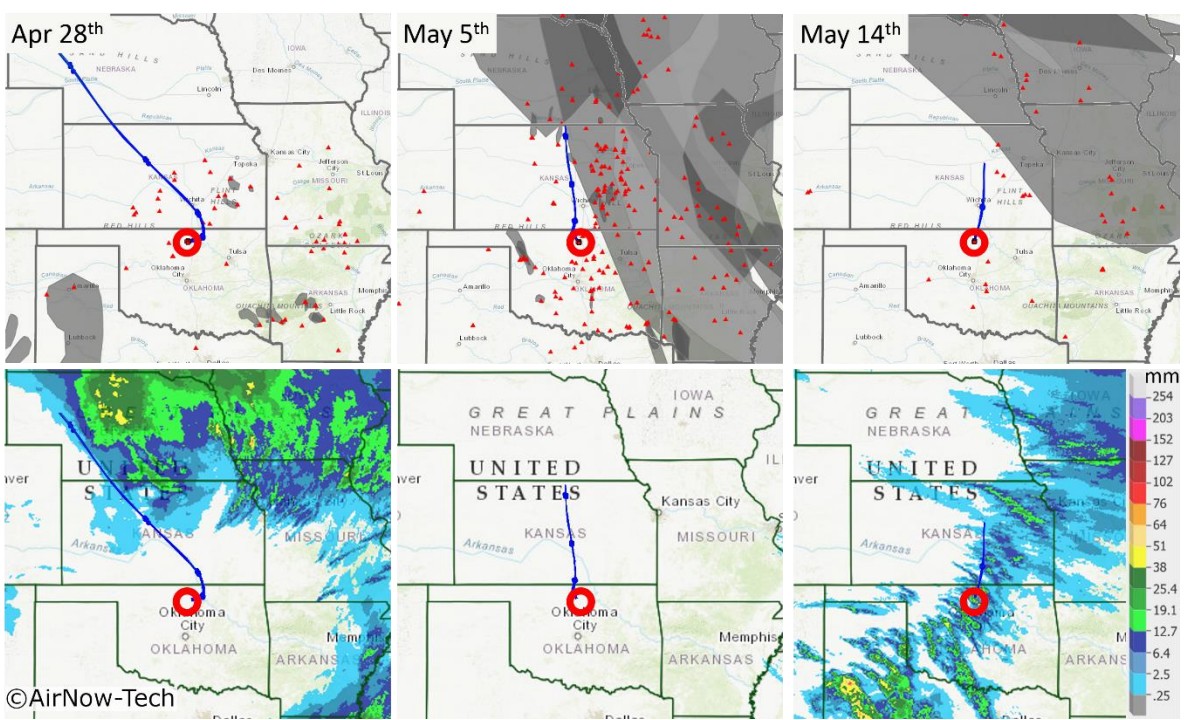

**Figure 4: Smoke, fire, and precipitation data along with HYSPLIT back trajectories for three sample dates. The red circle represents the sampling site while the small red triangles represent fires. The gray overlays seen in the top row represent detected smoke particles (overlapping smoke plumes are shown in darker shades of gray). The bottom row shows the 24-hour average precipitation amount over the sampling date. The top row maps were obtained using the AirNow-Tech navigator using the Hazard Mapping System smoke product from NOAA, Source: U.S. EPA AirNow-Tech (SonomaTechnologiesInc., 2019). HYSPLIT trajectories for April 28th and May 5th are for 24 hours. The May 14th back trajectory was truncated at 10 hours due to a rain event with significant precipitation scavenging. Precipitation maps were made using the NWS AHPS (NWS, 2019).**

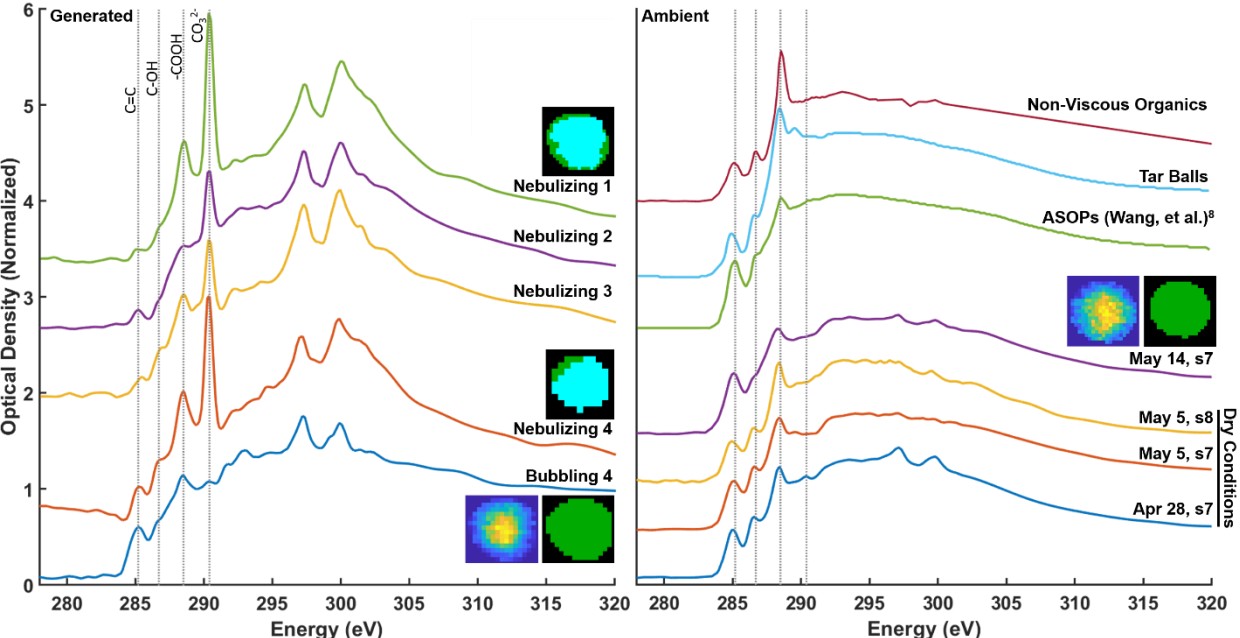

**Figure 5: Comparison of NEXAFS spectra between laboratory generated ASOP proxies (left) and four ambient ASOP samples (right). Inset carbon speciation maps are shown for representative particles with green representing organic dominant regions and teal representing inorganic dominant regions. TCA images, like the ones shown in Fig. 2 are shown as well. Only one image is shown for the ambient samples as they all look similar. The ASOP proxy spectra are show for particles generated by nebulizing aquatic samples collected at the site and by bubbling $N_2$ gas through one of the aquatic samples.**

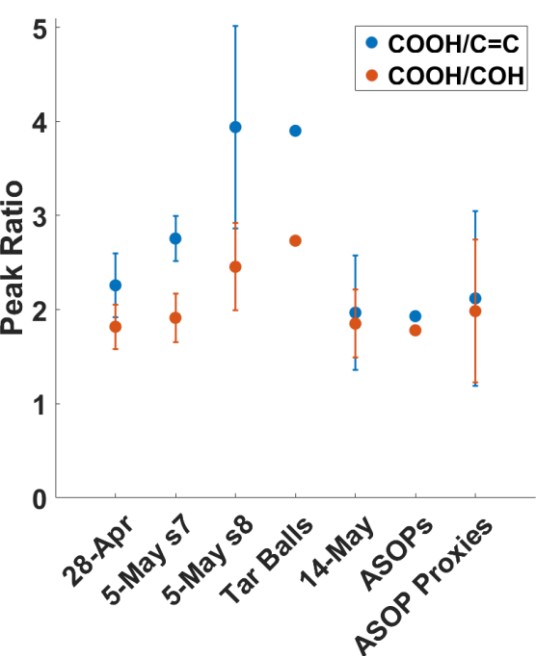

**Figure 6: Plot of peak ratios for: (blue dots) carboxylic acid peak at 288.6 eV and the carbon double bond peak at 285.3 eV and (red dots) carboxylic acid peak and the COH peak at 286.7 eV. ASOP Proxies refers to particles generated via bubbling through the aquatic sample of SOM.**