# Peer review of "Optical properties and composition of viscous organic particles found in the Southern Great Plains"

_Atmospheric Chemistry and Physics, 2020_

## Referee Comment (RC1) · Anonymous Referee #1 · 14 May 2020

The manuscript "Optical properties and composition of viscous organic particles found in the Southern Great Plains" presents the observation results of HVOP during April and May 2016. The works focused to discriminate between ASOP and TB in the HVOP. The studies provide different methods to characterize the ASOP. However, my primary concern is that the authors did not make it clear the contribution of ASOP to the BrC and how important the ASOP is in the atmosphere. Major comments 1. The experimental section "a portion of one SOM sample was used to generate particles by bubbling N2 gas through the liquid using a fritted glass bubbler and then collected on substrates to mimic the hypothesized mechanism of ASOP formation." The authors should give the detail information about the bubbling simulation processes and parameters. 2.

[Figure]

"where ASOP comprised up to 80% of the particles by number." The impactor can only provide the mass concentration. But the ASOP fraction was obtained based on particle number. Why? What is used to determine the ASOP number? 3. Precipitation results in the emissions of ASOP, but the AAE did not present an obvious increase during the rain events. Why? 4. In Fig. 1, the AAE did not correlate with the particle number. Why? 5. "ASOP may make up a small fraction of the HVOP seen during this sampling period." Indicating that most of the HVOP comes from the anthropogenic sources, such as coal-fired power plant, oil refinery and natural gas power plants. My concern is that why we should focus on the ASOP, as it may have few impacts on aerosol radiative forcing and climate effects. 6. "Our results support the bubble bursting mechanism of particle generation during rainfall resulting in the ejection of soil organics into the atmosphere." But the paper did not discuss how the ASOP ejected from soil to atmosphere during rainfall. The authors should include this parts in the paper.

---

## Referee Comment (RC2) · Anonymous Referee #2 · 9 Jun 2020

Fraund et al. conducted observations of characteristics of high viscosity organic particles (HVOP) apportioned as either airborne soil organic particles (ASOP) or tar balls (TB) during the HI-SCALE campaign. Formation mechanisms and properties of ASOP or TB have been paid much attention in recent years in the atmospheric chemistry community because of their role in climate forcing. There are few studies observing ASOP and TB in the same campaign and comparing their differences. As the authors pointed out that it is challenging to differentiate between ASOP and TB, this study is timely and I recommend the publication after the following comments can be addressed.

Major comments:

[Figure]

(1) The analysis of the three particular periods (April 28th, May 5th and 14th) with elevated AAE is reasonable and comprehensive. However, I am curious how these ASOP or TB went through chemical and physical changes after they were emitted? For example, the authors expected that HVOP in the May 14th samples would be present for at least 10 hours after the rain event (Line 295). Why the AAE gradually increased after the rain event and reached the peak $\sim$10 hours later (Fig. 1)? Did ASOP need this long time to be formed or accumulated? Would chemical ageing affect the particle composition during the 10 hours and make ASOP more viscous (Adachi et al., 2019)? What is the reason AAE decreased quickly after it reached the peak? Does it indicate the life time of HVOP is relatively short? Or they were transported to areas downwind? I understand it may not easy to clearly know the transformation of ASOP and TB, at least some discussions should be added.

(2) Line 322-325: The authors described the solubility of the non-spherical particles, which reminded me that what is the hygroscopicity of ASOP and TB? In recent years there are a lot of experiments measuring the viscosity of SOA formed from various precursors (summarized in Reid et al., 2018) at different relative humidity (RH). I see ASOP and TB are quite different from SOA but is it possible to measure the viscosity values of ASOP and TB varied with RH? Are ASOP and TB too small to measure their viscosity? As highly viscous secondary organic particles were observed frequently in ambient air (Virtanen et al., 2010 and other studies afterwards), I suggest at least add a few sentences describing the HVOP in this study is somehow different from those highly viscous secondary organic aerosol particles.

Minor comments:

(1) Line 38: delete "a" in "strong a climate . . .". Line 40: delete "aerosol" in "organic aerosol carbon". Line 78: should be Springston 2016 not 2011. (2) Line 119: Why the SOM aquatic samples were collected on May 17th? Why not choose 14th, same day that HVOP were sampled? (3) Line 186: The threshold (10 mm/hr) defining the rain events is different from 5 mm/hr described in Line 100. (4) Line 207: Is it certain

that an elevated AAE suggests the presence of spherical HVOP? I think better replace "can" to "may". (5) Line 217: There are two Moffet et al. together; please delete one. Also Line 234 for Veghte et al and Line 250 for Wang et al. (6) Table 1: what does IOP1 indicate? (7) Supplement: there are two "Figure S1". (8) Line 256: Cite Wang et al. 2016 after "lab-generated SOA particles". (9) Figure 1: It seems some grey vertical bars indicating BrC appearance do not have elevated AAE > 1.7, for instance, the night of May 1st. Why it is included in the "BrC events"?

References:

Adachi, K., Sedlacek, A. J., Kleinman, L., Springston, S. R., Wang, J., Chand, D., Hubbe, J. M., Shilling, J. E., Onasch, T. B., Kinase, T., Sakata, K., Takahashi, Y. and Buseck, P. R.: Spherical tarball particles form through rapid chemical and physical changes of organic matter in biomass-burning smoke, Proceedings of the National Academy of Sciences, 116, 19336-19341, 10.1073/pnas.1900129116, 2019.

Reid, J. P., Bertram, A. K., Topping, D. O., Laskin, A., Martin, S. T., Petters, M. D., Pope, F. D. and Rovelli, G.: The viscosity of atmospherically relevant organic particles, Nat. Commun., 9, 956, 10.1038/s41467-018-03027-z, 2018.

Virtanen, A., Joutsensaari, J., Koop, T., Kannosto, J., Yli-Pirilä, P., Leskinen, J., Mäkelä, J. M., Holopainen, J. K., Pöschl, U. and Kulmala, M.: An amorphous solid state of biogenic secondary organic aerosol particles, Nature, 467, 824-827, https://doi.org/10.1038/nature09455, 2010. Müller, L., Reinnig, M.-C., Warnke, J., and Hoffmann, Th.: Unambiguous identification of esters as oligomers in secondary organic aerosol formed from cyclohexene and cyclohexene/$\alpha$-pinene ozonolysis, Atmos. Chem. Phys., 8, 1423–1433, https://doi.org/10.5194/acp-8-1423-2008, 2008.
* * *

---

## Author Comment (AC1) · 7 Aug 2020

Please see the attached supplement for responses to all reviewer comments as well as the manuscript with tracked changes.

Please also note the supplement to this comment:
https://acp.copernicus.org/preprints/acp-2020-255/acp-2020-255-AC1-supplement.pdf

---

## Author Response (AR1)

**Responses to Reviewer 1**

The manuscript "Optical properties and composition of viscous organic particles found in the Southern Great Plains" presents the observation results of HVOP during April and May 2016. The works focused to discriminate between ASOP and TB in the HVOP. The studies provide different methods to characterize the ASOP. However, my primary concern is that the authors did not make it clear the contribution of ASOP to the BrC and how important the ASOP is in the atmosphere.

The authors would like to thank Reviewer #1 for their comments, suggestions, and time as well. We have made changes to the manuscript accordingly and have indicated below where these changes were made.

To address the primary concern about the magnitude of ASOP contribution to BrC and the atmosphere, Line 23 was added to the abstract:

> "one out of the three samples investigated had a significant number of ASOP while the other two samples contained TB."

Along with Line 33:

> "In addition, our results show that ASOP may only be atmospherically relevant during times when suitable emission conditions are met"

And Line 429 in the conclusion begins more discussion:

> "Of the three sampling days focused on in this study, only one indicated an appreciable number of ASOP present. Because the conditions necessary for ASOP emission depend both on soil properties and precipitation characteristics, the dominant source of HVOP will often be TB due to the frequency of biomass burning instances along with the large number of particles they emit. ASOP are likely to contribute to aerosol properties, optical and otherwise, only during short time periods where the emission conditions are met."

Major comments

1. The experimental section "a portion of one SOM sample was used to generate particles by bubbling N2 gas through the liquid using a fritted glass bubbler and then collected on substrates to mimic the hypothesized mechanism of ASOP formation." The authors should give the detail information about the bubbling simulation processes and parameters.

More detail added on Line 125 which now reads:

> "In addition, an aliquot of 30-40 mL of one SOM sample was used to generate particles by bubbling $N_2$ gas (at 8 L/min) through the liquid using a fritted glass bubbler. The $N_2$ gas, now carrying particles generated from the bursting of bubbles at the air-water interface was directed into a cascade impactor (Sioutas Personal Cascade Impactor #225-370, SKC) where particles were collected on a pre-loaded microscopy grid."

2. "where ASOP comprised up to 80% of the particles by number." The impactor can only provide the mass concentration. But the ASOP fraction was obtained based on particle number. Why? What is used to determine the ASOP number?

Rephrased to emphasize how the ASOP particles were identified on Line 124:

> "…to produce 300-500 nm diameter particles, where subsequent tilted SEM imaging revealed that ASOP comprised up to 80% of the particles by number."

3. Precipitation results in the emissions of ASOP, but the AAE did not present an obvious increase during the rain events. Why?

Added some additional discussion on Line 200 which now reads:

> "While ASOP emissions are expected during rain events (Joung and Buie, 2015), precipitation scavenging is also occurring. The net effect of these two competing processes likely depends on many environmental conditions and is not yet clear. During a rain event, ASOP will contribute to an increased AAE but because AAE is a bulk optical property the presence of other absorbing aerosols like black carbon or the washout of mineral dust (large particles with a high AAE) can dampen the effect that rain events have on AAE (Bergstrom et al., 2007)."

4. In Fig. 1, the AAE did not correlate with the particle number. Why?

To expand on this Line 257 now reads:

"Even though elevated HVOP numbers were identified on a number of days, Fig. 1 shows that there is no clear relationship between the particle concentration and AAE. One reason for this is the presence of other absorbing or non-absorbing aerosols which will increase the measured particle concentration while affecting the AAE differently than HVOP are expected to."

And Line 263 now reads:

"This correlation also uses only aerosols impacted onto MOUDI stages 7 and 8, whose combined aerodynamic size range (0.18 – 0.56 μm) covers much of the ambient aerosol surface area distribution (Seinfeld and Pandis, 2006). Because optical properties like AAE will be most sensitive to changes in this size range, we exclude particles which are counted in the overall particle concentration but contribute less to bulk optical properties."  .

5. "ASOP may make up a small fraction of the HVOP seen during this sampling period." Indicating that most of the HVOP comes from the anthropogenic sources, such as coal-fired power plant, oil refinery and natural gas power plants. My concern is that why we should focus on the ASOP, as it may have few impacts on aerosol radiative forcing and climate effects.

To address concerns about the importance of ASOP see additions to abstract and conclusion mentioned above.

More clarification was added to better define the sampling period this paragraph was referring to. Line 296 now reads:

"Although it is possible ASOP may make up a fraction of the HVOP seen during the April 28th sample, immediately east of the sampling site are a coal-fired power plant, an oil refinery, and natural gas power plants which could possibly contribute to the HVOP seen on this day."

6. "Our results support the bubble bursting mechanism of particle generation during rainfall resulting in the ejection of soil organics into the atmosphere." But the paper did not discuss how the ASOP ejected from soil to atmosphere during rainfall. The authors should include this parts in the paper.

The aerosol generation mechanism was expanded upon in the introduction at Line 60 which now reads:

"Briefly, as falling water droplets make contact with the porous soil surface, air from within the soil is trapped beneath the resulting water layer. Bubbles form as the droplet sinks and mixes with the soil. These bubbles then rise to the air-water interface bringing dissolved soil organics with them where a cavity forms and then ruptures producing aerosols containing compounds entrained from the soil (Joung and Buie, 2015)."

**Responses to Reviewer 2**

Fraund et al. conducted observations of characteristics of high viscosity organic particles (HVOP) apportioned as either airborne soil organic particles (ASOP) or tar balls (TB) during the HI-SCALE campaign. Formation mechanisms and properties of ASOP or TB have been paid much attention in recent years in the atmospheric chemistry community because of their role in climate forcing. There are few studies observing ASOP and TB in the same campaign and comparing their differences. As the authors pointed out that it is challenging to differentiate between ASOP and TB, this study is timely and I recommend the publication after the following comments can be addressed.

The authors would like to tank Reviewer #2 for their detailed comments, suggestions, and time as well as for recommending this manuscript for publication. We have made changes to the manuscript accordingly and have indicated below where these changes were made.

Major comments:

(1) The analysis of the three particular periods (April 28th, May 5th and 14th) with elevated AAE is reasonable and comprehensive. However, I am curious how these ASOP or TB went through chemical and physical changes after they were emitted? For example, the authors expected that HVOP in the May 14th samples would be present for at least 10 hours after the rain event (Line 295). Why the AAE gradually increased after the rain event and reached the peak _10 hours later (Fig. 1)? Did ASOP need this long time to be formed or accumulated? Would chemical ageing affect the particle composition during the 10 hours and make ASOP more viscous (Adachi et al., 2019)? What is the reason AAE decreased quickly after it reached the peak? Does it indicate the life time of HVOP is relatively short? Or they were transported to areas downwind? I understand it may not easy to clearly know the transformation of ASOP and TB, at least some discussions should be added.

Discussion on the aging of TB and ASOP was added to Line 350, which now reads:

"The chemical composition of TB has been reported in literature, with fresh TB being comprised of various biomass tar products with substantial degree of aromaticity. The nonpolar products were the most strongly associated with the wavelength dependence in absorption seen in TB and were found in greater number in fresh TB (Li et al., 2019). Photochemical oxidation in the presence of O3 or OH radicals was shown to bleach this wavelength dependence after about 3.5 days (Sumlin et al., 2017). Photooxidation was suggested to break the network of the conjugated double bonds of the TB chromophores, resulting in more oxygenated (carbonyl substituted) products which concentrated on the surface of the TB (Hand et al., 2005;Tóth et al., 2014). ASOP are comprised, in part, of high molecular weight humic-like substances. These compounds contain multiple conjugated ring systems which likely serve as chromophores in a similar way to the tar materials in TB (Kumada, 1955). The extensive conjugated systems may be driving the enhanced C=C peak in Fig. 5 for ASOP associated spectra. Also, because humic-like substances in ASOP are substituted by many different functional groups, rather than the nonpolar components of TB, more reactive sites may be available for oxidation. This may lead to faster atmospheric processing and bleaching of ASOP wavelength dependent absorption. This aging could also serve to increase the viscosity of ASOP in the same way that it does to TB (Adachi et al., 2019)."

Extra discussion on the 10-hour delay seen in the April 14th sample was also added to Line 321:

"It is also possible that emission conditions for ASOP (soil and rainfall characteristics) were ideal somewhere along the airmass trajectory. This would have the effect of bringing in a high concentration of ASOP after the aerosol plume traveled. In addition, production of viscous SOA particles was likely taking place at the same time. The peak in AAE seen during the May 14th sampling period occurs as the sun is rising and ozone concentration rapidly increases (this also coincides with an increase in particle concentration). Although viscous SOA particles would contribute to the AAE, they are formed <100 nm in size, limiting their effect on optical properties. The peak in AAE here, and in the other cases, drops down quickly due to the particle-laden airmass moving away from the sampling site."

(2) Line 322-325: The authors described the solubility of the non-spherical particles, which reminded me that what is the hygroscopicity of ASOP and TB? In recent years there are a lot of experiments measuring the viscosity of SOA formed from various precursors (summarized in Reid et al., 2018) at different relative humidity (RH). I see ASOP and TB are quite different from SOA but is it possible to measure the viscosity values of ASOP and TB varied with RH? Are ASOP and TB too small to measure their viscosity? As highly viscous secondary organic particles were observed frequently in ambient air (Virtanen et al., 2010 and other studies afterwards), I suggest at least add a few sentences describing the HVOP in this study is somehow different from those highly viscous secondary organic aerosol particles.

A comparison between viscous SOA and ASOP/TB is made on Line 371:

"Viscous SOA have been observed previously, under both laboratory and ambient conditions (Virtanen et al., 2010). The HVOP discussed in the current work differ from these viscous SOA by virtue of their mode of formation with both TB and ASOP being comprised of larger organic compounds. Furthermore, the particles themselves are larger. Viscous SOA are formed and observed as much smaller particles (<100 nm) then TB and ASOP (300 – 700 nm)."

A discussion of ASOP/TB hygroscopicity was also added to Line 362, now reading:

"Another factor differentiating their aging processes could be their hygroscopicity. Reports have shown TB change very little in morphology even when cycled from 0 to 100% RH, having a growth factor of ~1.09 (Adachi and Buseck, 2011;Semeniuk et al., 2007). However, because ASOP are formed via dissolved SOM that is ejected during precipitation, they are expected to be more hygroscopic. Indeed, Wang (2016) showed the results of RH cycling in supplemental figures and found a growth factor of ~1.15 at 85% RH along with droplet activation at 98% RH."

Minor comments:
(1) Line 38: delete "a" in "strong a climate : : :". Deleted "a"
Line 40: delete "aerosol" in "organic aerosol carbon". Deleted "aerosol"
Line 78: should be Springston 2016 not 2011. Reference fixed

(2) Line 119: Why the SOM aquatic samples were collected on May 17th? Why not choose 14th, same day that HVOP were sampled? Added to Line 122: "..., prior to offline analysis of microscopy samples"

(3) Line 186: The threshold (10 mm/hr) defining the rain events is different from 5 mm/hr described in Line 100. All threshold values changed to 10 mm/hr

(4) Line 207: Is it certainthat an elevated AAE suggests the presence of spherical HVOP? I think better replace "can" to "may". Changed "can" to "may"

(5) Line 217: There are two Moffet et al. together; please delete one. Also Line 234 for Veghte et al and Line 250 for Wang et al.
    Reworded Line 227 to read "Following the procedure described in Moffet (2010) for chemical speciation maps, each… or as a region with high C=C bonding (Moffet et al., 2010)."
    Repeated instances of Veghte et al and Wang et al were removed.

(6) Table 1: what does IOP1 indicate?
    Table 1 caption now reads: "Ambient sampling information…"

(7) Supplement: there are two "Figure S1".
    The second supplemental figure was renamed "Figure S2"

(8) Line 256: Cite Wang et al. 2016 after "lab-generated SOA particles".
    Inserted Wang et al., 2016 citation

(9) Figure 1: It seems some grey vertical bars indicating BrC appearance do not have elevated AAE > 1.7, for instance, the night of May 1st. Why it is included in the "BrC events"?

Modified Figure 1 caption to read "…Grey vertical bars indicate sampling periods where microscopy samples were collected… Arrows indicate the three sampling periods investigated in the current work where BrC particles were expected."

[revised manuscript text omitted]

**Figure S3. Hybrid single particle Lagrangian integrated trajectory (HYSPLIT) back trajectories for three sample dates in 2016 at different starting altitudes. The starting location (represented by the black star) is the Lamont, OK central ARM facility at 36.60 N and 97.49 W. The April 28th and May 5th plots are 24 hour trajectories while the May 14th plots are 10 hour trajectories. Different colored trajectories within a single plot represent new trajectories every 3 hours after the initial starting time. Archived data was taken from the North American Mesoscale (NAM) sigma-pressure hybrid model.**